

# Association between *VDR* gene *FokI* polymorphism and renal function in patients with IgA nephropathy

Man-Qiu Mo\*, Ling Pan\*, Lin Tan, Ling Jiang, Yong-Qing Pan, Fu-Ji Li, Zhen-Hua Yang and Yun-Hua Liao

Department of Nephrology, the First Affiliated Hospital of Guangxi Medical University, Nanning, China
\* These authors contributed equally to this work.

## ABSTRACT

**Background:** Studies have shown that the occurrence and development of IgA nephropathy (IgAN) are genetically susceptible, but the relationship between vitamin D receptor (*VDR*) gene polymorphisms and renal function in IgAN patients is unclear.

**Methods:** We investigated the relationship between *VDR FokI* (rs2228570) single nucleotide polymorphism (SNP) and renal function and related clinicopathologic parameters in IgAN patients. Clinical and pathological data of 282 IgAN patients treated at the First Affiliated Hospital of Guangxi Medical University were collected, and *FokI* genotypes were determined by PCR and direct sequencing. Patients were divided into the renal dysfunction group and normal renal function (control) group by estimated glomerular filtration rate (eGFR) and serum creatinine level.

**Results:** Frequencies of TT genotype and T allele in the renal dysfunction group were higher than those of the control group. Blood urea nitrogen, serum phosphorus (P), proportions of mesangial cell proliferation, interstitial fibrosis/tubular atrophy and crescents in T allele carriers were higher than those in non-T allele carriers, while eGFR and 25-Hydroxyvitamin D3 were lower in T allele carriers than non-T allele carriers. Multiple linear regression analysis showed that eGFR was affected by *FokI* genotypes in IgAN patients. Logistics regression analysis showed that middle and elderly age, elevated P, intact parathyroid hormone and TT genotype were independent risk factors for renal dysfunction in IgAN patients; the odds ratio of carrying the TT genotype was as high as 84.77 (*P* < 0.05 for all).

**Conclusions:** IgA nephropathy patients carrying the *VDR FokI* TT genotype have an increased risk of renal dysfunction. *VDR FokI* SNP is closely related to renal function, calcium-phosphate metabolism, and related pathological damage in IgAN patients.

## INTRODUCTION

As one of the most common primary glomerular diseases, IgA nephropathy (IgAN) results in a high incidence of renal dysfunction, and more than 40% of patients progress to

Corresponding authors
Zhen-Hua Yang,
593456108@qq.com
Yun-Hua Liao,
yunhualiao1962@163.com

end-stage renal disease (ESRD) within 20 years (*Li et al., 2015*). Many studies have shown that genetic cause is one of the primary factors affecting IgAN (*Magistroni et al., 2015*). Recently, genome-wide association studies (GWAS) have identified 20 genetic susceptibility genes of IgAN, which are closely related to the occurrence, progression and prognosis of IgAN.

The vitamin D receptor (*VDR*) gene is a crucial mediator of active vitamin D biological activity. Active vitamin D binds to VDR to regulate target gene transcription, thus realizing biological functions such as regulation of calcium-phosphorus metabolism, immunity, and inflammatory state (*Christensen et al., 2013*). *VDR* polymorphisms are associated with occurrence and prognosis of certain diseases, such as chronic kidney disease (CKD), ischemic stroke, and malignant tumors (*Cho et al., 2018*; *Prabhakar et al., 2015*; *Santoro et al., 2013*). *FokI* is one locus located in the initiation codon region of *VDR*, which is the only functional locus known to affect VDR protein peptide chain structure (*Hu et al., 2017*). Previous studies have indicated that *FokI* polymorphism was associated with diabetic nephropathy and lupus nephritis, which increased the susceptibility of chronic renal failure (CRF) (*Imam et al., 2017*; *Razi et al., 2019*). However, the relationship between *VDR* polymorphisms and renal function of IgAN patients is unclear. Hence, we investigated the association between *VDR FokI* single nucleotide polymorphism (SNP) and renal function and related clinicopathological damage of IgAN.

## MATERIALS AND METHODS

### Subjects

All subjects were patients treated in the Department of Nephrology at the First Affiliated Hospital of Guangxi Medical University from August 2014 to December 2016, who were diagnosed with IgAN by renal biopsy. Inclusion criteria were patients who met diagnostic criteria of the 2014 edition of the IgAN Guidelines issued by the Japanese Society of Nephrology (*Yuzawa et al., 2016*) and age $\geq$16 years. Exclusion criteria were secondary IgAN (secondary to autoimmune diseases such as allergic purpura, systemic lupus erythematosus, dry syndrome, arthritis and psoriasis or secondary to hepatobiliary and gastrointestinal diseases, respiratory diseases, viral infections, and tumors) (*Saha et al., 2018*), patients with severe liver failure, patients with acute cardiovascular diseases or acute cerebrovascular diseases and patients with a history of medications such as steroid hormones, immunosuppressants, and active vitamin D preparations/analogs for the past 3 months. The study was approved by the Ethics Committee of the First Affiliated Hospital of Guangxi Medical University (Approval Number: 2019KY-E-006), and all participants were aware of the purpose of this study and provided written informed consent. Ultimately, 282 subjects were included in this study.

### Clinical data

Basic information of subjects was collected by questionnaires. Height, weight and vital signs were recorded. Body mass index (BMI) was calculated by the following formula: BMI = Weight (kg)/Height$^2$ (m$^2$). Pulse pressure was calculated by the following formula: Pulse pressure = Systolic blood pressure (SBP) – Diastolic blood pressure (DBP).

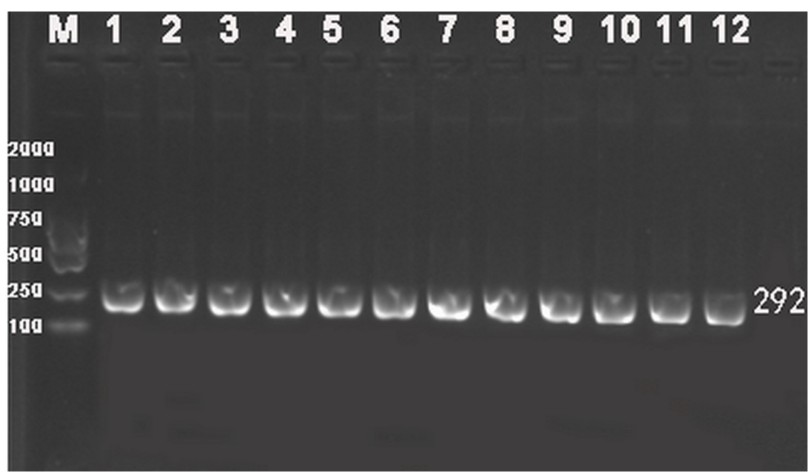

**Figure 1 Representative electrophoresis gel of PCR for *VDR FokI* (rs2228570) polymorphism.** Lane M, 100-bp marker ladder; Lanes 1–12, PCR products from 12 different DNA samples (292 bp).

Peripheral venous blood (four mL) was obtained from each participant and used to examine liver and kidney function, electrolytes, iPTH, 25-Hydroxyvitamin D3 (25(OH)D3) and 25-Hydroxyvitamin D2 (25(OH)D2). We also collected 24-h urine volume for urine protein quantitation. Morning urine samples (approximately five mL) were used for routine urine testing.

## Clinical grouping

Estimated glomerular filtration rate (eGFR) was calculated using the CKD Epidemiology Collaboration equation according to serum creatinine (Scr), sex, and age (*Levey et al., 2017*). eGFR <60 mL/min/1.73 m$^2$ and/or Scr >104 µmmol/L in males or Scr >84 µmmol/L in females was considered to indicate renal dysfunction. Subjects were then divided into the renal dysfunction group ($n = 156$) and the control group (normal renal function group, $n = 126$) for analyses.

## DNA amplification, sequencing, and genotyping

Genomic DNA was extracted from peripheral blood leukocytes using the phenol-chloroform method (*Nan et al., 2016*). Based on gene sequences provided by the National Center for Biotechnology, specific primers were devised and aligned by Primer 5.0 software (Premier Company, North York, Canada). The primer pair rs2228570-3F: 5′-TGGGT GGCACCAAGGATG-3′ and rs2228570-3R: 5′-CCTTCATGGAAACACCTTGC-3′ was synthesized by Shanghai Sangon Biological Engineering Technology & Services Co., Ltd., China. PCR was performed in 40 µL volume reactions comprising 20 µL of master mix, 17 µL of water, one µL of each upstream and downstream primer and one µL of DNA template. PCR cycle conditions were as follows: initial denaturing at 94 °C for 5 min, followed by 35 cycles of denaturing at 94 °C for 30 s, annealing at 55 °C for 30 s and elongation at 72 °C for 30 s. Amplification was completed with a final extension at 72 °C for 5 min. PCR products were visualized on 2.0% agarose gels (Fig. 1). *FokI* genotype was

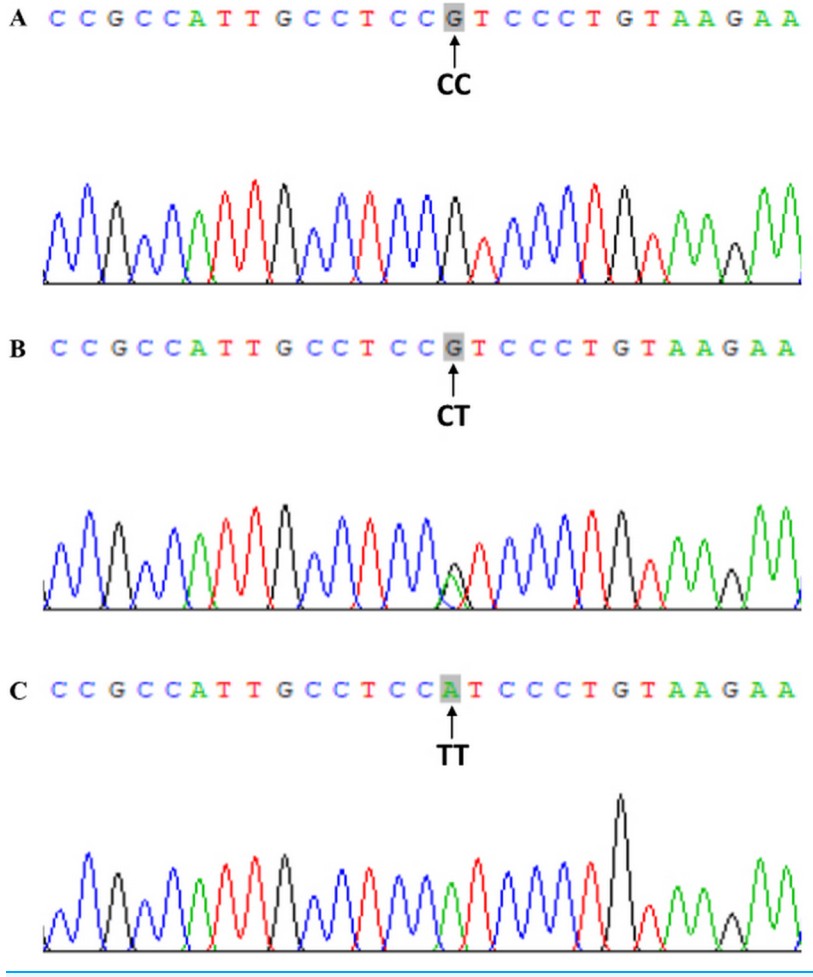

**Figure 2 Chromatograms of direct sequencing for *VDR FokI* (rs2228570) polymorphism.** (A) CC, CC genotype; (B) CT, CT genotype; (C) TT, TT genotype.

determined by direct sequencing of all PCR products and performed by Shanghai Sangon Bioengineering Company (partial nucleotide sequences of CC, CT, and TT genotypes are shown in Fig. 2).

## Diagnosis and staging criteria of renal pathology

IgA nephropathy was diagnosed according to renal pathology. IgAN is characterized by proliferation of mesangial cells, and IgA is dominant or co-dominant in glomerular deposition, which is usually accompanied by complement C3 and varying amounts of IgG and/or IgM (*Kiryluk & Novak, 2014*).

Renal pathology staging was based on the Oxford classification and crescent scores. Scores were defined as follows: M1, >50% of the mesangial area consisting of more than three mesangial cells, otherwise scored as M0. E1, narrowed glomerular capillary cavities due to cell proliferation, otherwise scored as E0. S1, any degree of injury in capillary fistula excluding injury to the entire glomerulus or adhesion, otherwise scored as S0. Interstitial fibrosis/tubular atrophy (IFTA) was defined as follows: T0, 0–25%; T1, 26–50%;
and T2, >50%. Crescents were scored as follows: C0, no crescent; C1, crescents in <25% of the glomerulus; and C2, crescents in ≥25% of the glomerulus (*Trimarchi et al., 2017*).

## Statistical analysis

Statistical analyses were performed using SPSS version 21.0 (SPSS, Chicago, IL, USA). Data with a normal distribution were expressed as mean ± standard deviation (SD) or percentages. Data with a non-normal distribution were presented as the median and quartile. The Chi-squared test was used to determine Hardy–Weinberg equilibrium (HWE). The *t*-test and Chi-squared test were used to determine differences of indicators between two groups. Comparison of clinical indicators among different genotypes was analyzed by variance analysis; distribution of pathological staging among different genotypes was compared by the Chi-squared test. Multiple linear regression analysis and binary logistic regression analysis were used to analyze risk factors associated with renal dysfunction in IgAN. $P < 0.05$ was considered to be statistically significant in all analyses.

## RESULTS

We enrolled 282 IgAN patients; the sex ratio (male:female) was 1:1.19 (129/153), mean age (± SD) was 33.67 ± 10.75 years and 156 patients had renal dysfunction (55.32%). There were no significant differences in age and sex between renal dysfunction and control groups. Compared with the control group, SBP, DBP, pulse pressure, blood urea nitrogen (BUN), Scr, uric acid (UA), 24-h urine protein quantitation (24hUPro) and intact parathyroid hormone (iPTH) levels of the renal dysfunction group were higher, while eGFR, hemoglobin (Hb), 25(OH)D3 and 25(OH)D2 were lower ($P < 0.05$ for all). Detailed information is shown in Table 1.

Distribution of *FokI* genotype and allele frequencies was consistent with HWE in enrolled IgAN patients ($P > 0.05$, Table 1). Frequencies of CC, CT, and TT genotypes were 19.15%, 50.35% and 30.50%, and frequencies of C and T alleles were 44.33% and 55.67%, respectively. Frequencies of TT genotype and T allele in the renal dysfunction group were significantly higher than those of the control group ($P < 0.0001$).

Comparison of clinical and pathological indicators between different genotypes is shown in Table 2. There were significant differences in BUN, UA, phosphorus (P), eGFR, 25(OH)D3, 25(OH)D2, extent of mesangial cell hyperplasia, capillary intravascular hyperplasia, IFTA, and crescents among different genotypes. BUN and P of T allele carriers (TT+CT) were higher than those of non-T allele (CC) carriers, while eGFR, 25(OH)D3 and 25(OH)D2 levels of T allele carriers were lower than those of non-T allele carriers. Furthermore, proportions of M1, T2, and C2 of T allele carriers were higher than those of non-T allele carriers ($P < 0.05$ for all).

Multiple linear regression analysis showed that eGFR was closely associated with *FokI* genotypes ($P < 0.05$, Table 3). Additionally, eGFR was also affected by various factors such as age, BMI, Hb, 25(OH)D3 and pathological stage ($P < 0.05$ for all). Logistic regression analysis showed that TT genotype may be a significant factor of renal dysfunction in IgAN after adjusting for sex, age, P, iPTH, and other factors (odds ratio (OR) = 84.77, $P < 0.0001$), and the risk of renal dysfunction in TT genotype carriers

**Table 1 Clinical characteristics and genotype and allele frequencies of IgAN patients.**

| Parameters | Total | Renal dysfunction | Control | $t(X^2/Z)$ | $P$ |
|---|---|---|---|---|---|
| Number | 282 | 156 (55.32%) | 126 (44.68%) | | |
| Male/female | 129/153 | 69/87 | 60/66 | 2.406 | 0.121 |
| Age (year) | 33.67 ± 10.75 | 34.58 ± 9.89 | 32.55 ± 11.66 | 1.585 | 0.114 |
| BMI (kg/m$^2$) | 22.06 ± 3.39 | 21.75 ± 3.42 | 22.44 ± 3.32 | −1.707 | 0.089 |
| SBP (mmHg) | 132.5 ± 17.97 | 140.85 ± 13.89 | 122.36 ± 17.20 | 9.764 | <0.0001 |
| DBP (mmHg) | 81.33 ± 12.69 | 86.64 ± 11.19 | 74.76 ± 11.32 | 8.816 | <0.0001 |
| Pulse pressure (mmHg) | 51.25 ± 10.92 | 54.21 ± 10.56 | 47.60 ± 10.28 | 5.289 | <0.0001 |
| eGFR (mL/min/1.73 m$^2$) | 76.49 ± 35.91 | 53.63 ± 21.11 | 104.79 ± 29.79 | −16.260 | <0.0001 |
| Hb (g/L) | 120.4 ± 20.58 | 116.86 ± 23.46 | 124.79 ± 15.30 | −3.414 | 0.001 |
| BUN (mmol/L) | 5.87 ± 2.46 | 6.74 ± 2.76 | 4.78 ± 1.41 | 7.687 | <0.0001 |
| Scr (μmol/L) | 116.2 ± 99.27 | 153.2 ± 120.95 | 70.50 ± 13.88 | 8.476 | <0.0001 |
| UA (μmol/L) | 400.06 ± 112.81 | 452.85 ± 97.83 | 334.71 ± 94.66 | 10.264 | <0.0001 |
| Ca (mmol/L) | 2.19 ± 0.14 | 2.18 ± 0.13 | 2.21 ± 0.16 | −1.215 | 0.226 |
| P (mmol/L) | 1.36 ± 0.48 | 1.34 ± 0.39 | 1.38 ± 0.58 | −0.665 | 0.507 |
| iPTH (mmol/L) | 51.45 (16.26) | 55.76 (25.72) | 41.75 (14.67) | −6.891 | <0.0001 |
| 24hUPro (g/d) | 1.71 ± 1.54 | 2.08 ± 1.20 | 1.26 ± 1.78 | 4.426 | <0.0001 |
| 25(OH)D3 (nmol/L) | 41.89 ± 19.61 | 38.91 ± 18.39 | 45.68 ± 20.52 | −2.759 | 0.006 |
| 25(OH)D2 (nmol/L) | 4.39 ± 2.04 | 4.08 ± 1.93 | 4.77 ± 2.12 | −2.693 | 0.008 |
| Genotypes | | | | | |
| CC (n(%)) | 54 (19.15) | 15 (9.62) | 39 (30.95) | | |
| CT (n(%)) | 142 (50.35) | 76 (48.72) | 66 (52.38) | | |
| TT (n(%)) | 86 (30.50) | 65 (41.66) | 21 (16.67) | 31.042 | <0.0001 |
| Alleles | | | | HWE(P) | 0.734 |
| C (n(%)) | 250 (44.33) | 106 (33.97) | 144 (57.14) | | |
| T (n(%)) | 314 (55.67) | 206 (66.03) | 108 (42.86) | 30.322 | <0.0001 |

Note:
25(OH)D2, 25-Hydroxyvitamin D2; 25(OH)D3, 25-Hydroxyvitamin D3; 24hUpro, quantitative 24-h urinary protein; BMI, body mass index; BUN, blood urea nitrogen; Ca, calcium; DBP, diastolic blood pressure; eGFR, estimated glomerular filtration rate; Hb, hemoglobin; iPTH, intact parathyroid hormone; P, phosphorus; SBP, systolic blood pressure; Scr, serum creatinine; UA, uric acid.

was 84.77-fold higher than that in CC genotype carriers with IgAN. Furthermore, age >40 years (OR = 9.60, $P$ = 0.005), serum P (OR = 15.68, $P$ = 0.001) and iPTH (OR = 1.13, $P$ < 0.0001) were independent risk factors of renal dysfunction in IgAN patients. Detailed information is shown in Table 4.

## DISCUSSION

IgA nephropathy accounts for 40–47.2% of primary glomerular nephritis in China and is one of the most common primary glomerular diseases worldwide and an important cause of CRF (*Wang et al., 2015*; *Wyatt & Julian, 2013*). Our results indicated that 55.32% of IgAN patients have renal dysfunction, which is similar to the proportion of primary IgAN patients with renal failure in India (47%) (*Chowdry et al., 2018*). We also found that the *VDR* gene was closely related to elevated serum iPTH and P, and elevated iPTH and P were independent risk factors for renal dysfunction in IgAN. Similar to our results,

**Table 2 Comparison of clinical and pathological characteristics between IgAN patients by genotypes.**

| Parameters | CC | CT | TT | $F(X^2)$ | P |
|---|---|---|---|---|---|
| SBP (mmHg) | 134.22 ± 22.15 | 129.23 ± 17.27 | 137.09 ± 15.03 | 5.579 | 0.004 |
| DBP (mmHg) | 79.78 ± 12.23 | 81.71 ± 13.07 | 81.69 ± 12.41 | 0.500 | 0.607 |
| Pulse pressure (mmHg) | 54.44 ± 11.80 | 47.52 ± 9.55 | 55.41 ± 10.44 | 18.958 | <0.0001 |
| BUN (mmol/L) | 5.50 ± 2.41 | 6.31 ± 2.90 | 6.20 ± 2.14 | 3.343 | 0.037 |
| Scr (μmol/L) | 100.94 ± 49.37 | 111.01 ± 79.61 | 134.59 ± 141.45 | 2.330 | 0.099 |
| UA (μmol/L) | 370.44 ± 100.86 | 399.88 ± 132.50 | 418.97 ± 75.41 | 3.115 | 0.046 |
| eGFR (mL/min/1.73 m$^2$) | 81.30 ± 37.37 | 80.72 ± 36.13 | 66.47 ± 32.87 | 4.954 | 0.008 |
| 24hUPro (g/d) | 1.59 ± 1.96 | 1.81 ± 1.59 | 1.61 ± 1.08 | 0.697 | 0.499 |
| Ca (mmol/L) | 2.16 ± 0.19 | 2.20 ± 0.14 | 2.21 ± 0.09 | 1.879 | 0.155 |
| P (mmol/L) | 1.20 ± 0.32 | 1.44 ± 0.54 | 1.33 ± 0.44 | 5.254 | 0.006 |
| iPTH (mmol/L) | 50.73 ± 15.48 | 55.40 ± 28.37 | 58.16 ± 18.33 | 0.763 | 0.468 |
| 25(OH)D3 (nmol/L) | 47.88 ± 8.59 | 39.88 ± 21.56 | 39.61 ± 22.91 | 4.187 | 0.016 |
| 25(OH)D2 (nmol/L) | 5.02 ± 0.90 | 4.17 ± 2.23 | 4.16 ± 2.40 | 4.380 | 0.014 |
| M | | | | | |
| M0 (n(%)) | 39 (72.22) | 106 (74.65) | 36 (41.86) | | |
| M1 (n(%)) | 15 (27.78) | 36 (25.35) | 50 (58.14) | 26.925 | <0.0001 |
| E | | | | | |
| E0 (n(%)) | 42 (77.78) | 136 (95.77) | 83 (96.51) | | |
| E1 (n(%)) | 12 (22.22) | 6 (4.23) | 3 (3.49) | 21.198 | <0.0001 |
| S | | | | | |
| S0 (n(%)) | 24 (44.44) | 48 (33.80) | 24 (27.91) | | |
| S1 (n(%)) | 30 (55.56) | 94 (66.20) | 62 (72.09) | 4.048 | 0.132 |
| IFTA | | | | | |
| T0 (n(%)) | 39 (72.22) | 69 (48.59) | 77 (89.53) | | |
| T1 (n(%)) | 15 (27.78) | 64 (45.07) | 6 (6.98) | | |
| T2 (n(%)) | 0 (0) | 9 (6.34) | 3 (3.49) | 43.907 | <0.0001 |
| Crescent | | | | | |
| C0 (n(%)) | 45 (83.33) | 127 (89.44) | 83 (96.51) | | |
| C1 (n(%)) | 9 (16.67) | 12 (8.45) | 0 (0) | | |
| C2 (n(%)) | 0 (0) | 3 (2.11) | 3 (3.49) | 15.326 | 0.004 |

**Note:**
25(OH)D2, 25-Hydroxyvitamin D2; 25(OH)D3, 25-Hydroxyvitamin D3; 24hUpro, quantitative 24-h urinary protein; BUN, blood urea nitrogen; Ca, calcium; DBP, diastolic blood pressure; E, capillary hyperplasia; eGFR, estimated glomerular filtration rate; IFTA, interstitial fibrosis/tubular atrophy; iPTH, intact parathyroid hormone; M, mesangial cell proliferation; P, phosphorus; S, segmental glomerular sclerosis; SBP, systolic blood pressure; Scr, serum creatinine; UA, uric acid.

studies showed that secondary hyperparathyroidism (SHPT) and hyperphosphatemia not only affect renal progression but also increase the mortality of CKD, especially in ESRD (*Ritter & Slatopolsky, 2016*). Therefore, the *VDR* gene may be closely related to renal function and prognosis of IgAN patients.

Many studies have confirmed that IgAN has ethnic and regional differences and family aggregation tendencies (*Cox et al., 2017*; *Schena & Nistor, 2018*). Genetic causes are important mechanisms of IgAN occurrence and progression. A study from China

**Table 3 Relationship between eGFR and relative risk factors in IgAN patients.**

| Parameters | Risk factors | B | Std. error | Beta | t | P |
|---|---|---|---|---|---|---|
| eGFR (mL/min/1.73 m²) | BUN | −5.108 | 0.011 | −0.423 | −463.547 | <0.0001 |
| | SBP | −4.668 | 0.012 | −2.271 | −398.449 | <0.0001 |
| | UA | 0.088 | 0.000 | 0.292 | 206.940 | <0.0001 |
| | Age >40 | 94.215 | 0.180 | 0.885 | 524.790 | <0.0001 |
| | BMI | −4.498 | 0.036 | −0.447 | −124.938 | <0.0001 |
| | M | 24.518 | 0.073 | 0.369 | 335.974 | <0.0001 |
| | S | −22.264 | 0.072 | −0.277 | −311.333 | <0.0001 |
| | Alb | −7.961 | 0.013 | −1.234 | −628.707 | <0.0001 |
| | IFTA | 31.241 | 0.146 | 0.542 | 213.292 | <0.0001 |
| | 25(OH)D3 | 1.036 | 0.002 | 0.567 | 483.723 | <0.0001 |
| | P | −58.854 | 0.123 | −0.574 | −480.129 | <0.0001 |
| | iPTH | −1.946 | 0.004 | −0.862 | −553.134 | <0.0001 |
| | DBP | 5.674 | 0.013 | 1.860 | 439.998 | <0.0001 |
| | E | −10.019 | 0.109 | −0.062 | −91.619 | <0.0001 |
| | Hb | 0.065 | 0.002 | 0.041 | 38.457 | <0.0001 |
| | *FokI* genotypes | 9.469 | 0.067 | 0.181 | 142.337 | <0.0001 |
| | Crescent | −20.396 | 0.130 | −0.264 | −156.677 | <0.0001 |

Note:
25(OH)D3, 25-Hydroxyvitamin D3; Alb, albumin; BMI, body mass index; BUN, blood urea nitrogen; DBP, diastolic blood pressure; E, capillary hyperplasia; eGFR, estimated glomerular filtration rate; Hb, hemoglobin; IFTA, interstitial fibrosis/tubular atrophy; iPTH, intact parathyroid hormone; M, mesangial cell proliferation; P, phosphorus; S, segmental glomerulosclerosis; SBP, systolic blood pressure; UA, uric acid.

**Table 4 Analysis of risk factors related to renal dysfunction in IgAN patients.**

| Risk factors | Renal dysfunction | |
|---|---|---|
| | OR (95% CI) | P |
| *FokI* genotypes | | |
| CC | 1 | |
| CT | 2.79 [0.72–10.80] | 0.136 |
| TT | 84.77 [13.61–528.01] | <0.0001 |
| Age > 40 | 9.60 [1.98–46.47] | 0.005 |
| P | 15.68 [3.20–76.83] | 0.001 |
| iPTH | 1.13 [1.08–1.19] | <0.0001 |

Note:
iPTH, intact parathyroid hormone; P, phosphorus.

including 613 IgAN adult patients created the best predictive model of IgAN progression consisting of four loci (rs11150612, rs7634389, rs2412971, and rs2856717). According to the genetic risk score, the risk of disease progression in IgAN patients with moderate and high genetic risk was 2.12- and 3.61-fold higher than that of IgAN patients with low genetic risk, respectively (*Shi et al., 2018*). Our study indicated that the frequency of TT genotype in the renal dysfunction group was higher than that in the control group, and eGFR of T allele carriers was lower than that of non-T allele carriers. Thus, TT genotype is an independent risk factor of renal dysfunction. *VDR FokI* SNP is closely

associated with renal dysfunction in IgAN patients, which also supported the concept that genetic factors play an important role in IgAN renal function and progression. A previous study has shown that *VDR FokI* polymorphism is associated with risk of CRF in Asians (*Li et al., 2018*), which is also supported by our findings.

The *VDR* gene is located on the long arm of chromosome 12, with a total length of about 75 kb, and contains 11 exons and several introns. *VDR* polymorphisms can encode different proteins and cause various physiological effects. Studies have confirmed that *VDR* polymorphisms affect CKD occurrence and progression such as diabetic nephropathy, lupus nephritis, and hypertensive renal damage (*Mahto et al., 2018*; *Yang et al., 2017*). Our study also showed that *VDR FokI* is associated with renal dysfunction in IgAN patients. Renal function of IgAN may be influenced by *VDR* polymorphisms through the following reasons and mechanisms. First, *VDR* SNPs affect mRNA quality and stability by interfering with *VDR* mRNA expression and splicing, subsequently affecting VDR protein number and/or activity, and transactivates VDR protein and target gene(s) (*Karonova et al., 2018*). Second, *VDR* polymorphisms may affect CKD occurrence and development such as IgAN by inhibiting the anti-inflammatory activity of active vitamin D, protecting endothelial cells and promoting mesangial cell proliferation, podocyte loss, and tubulointerstitial fibrosis (*Yang et al., 2012*). The *FokI* locus is located on the transcription initiation site, and SNPs in this region can change the length of the amino acid sequence. When the genotype is T/T, the initiation codon in *VDR* is mutated from ATG to ACG, which results in loss of translation function and ultimately changes its biological functions such as proliferation and division (*Beysel et al., 2018*). This may be one of the potential molecular mechanisms by which *FokI* TT genotype carriers are more likely to suffer from renal dysfunction in IgAN patients. Ethnic differences in *VDR FokI* polymorphism and the influence of environmental, geographic, dietary, or occupational factors must be further elucidated.

Recently, with the development of molecular genetics and the completion of the Human Genome Project, greater attention has been paid to the pathogenesis of genetic factors in IgAN patients. To date, several GWAS have been carried out in IgAN patients of different ancestries, and at least 20 susceptibility loci have been identified, including human leukocyte antigen gene, complement factor H-related protein genes and tumor necrosis factor superfamily member 13 (*Han et al., 2016*; *Yang et al., 2018*; *Zhu et al., 2015*). However, the clinical application value is rarely found. Our study found that VDR gene polymorphism was associated with renal insufficiency of IgAN. Recent studies have shown that active vitamin D and its preparation can significantly alleviate renal dysfunction in CKD patients, reduce proteinuria, inhibit the secretion of parathyroid hormone and delay the progression to end-stage kidney disease and all-cause mortality (*Gluba-Brzozka et al., 2018*; *Melamed et al., 2018*). Furthermore, active vitamin D and its preparation are commonly administered to treat CKD-related mineral and bone metabolism (MBD), which improve the prognosis in CKD patients. Active vitamin D is not only an effective therapy for CKD but also a drug that can be safely administered for prolonged periods. As the necessary receptor of active vitamin D, VDR should have significant clinical applying prospects.

Our study found serum P of T allele carriers was higher than that of non-T allele carriers. *VDR FokI* polymorphism was shown to be associated with MBD (*Bouksila et al., 2018*). The T (or F) site of FokI protein is three amino acids shorter than the f (or C) site. Therefore, the affinity for ligand, mRNA stability and transport activity of F are higher than those of f (*Elias et al., 2018*), subsequently increasing intestinal calcium absorption, promoting calcium and phosphorus deposition in bone and affecting serum Ca and P levels. Simultaneously, SHPT can also inhibit phosphate reabsorption in renal proximal tubules. Studies have shown that *VDR* expression may be affected by calcium homeostasis and adequate calcium intake may offset the influence of genetic factors of *VDR* on bones (*Moran et al., 2015*). This may explain why we observed no significant difference in calcium levels between different genotypes. Our study also found that 25(OH)D3 of T allele carriers was lower than that of non-T allele carriers. Previous studies have shown that 25(OH)D deficiency is significantly associated with renal pathology severity and increases the risk of kidney progression (*Li et al., 2016*). Thus, the T allele of *VDR FokI* may be a susceptive allele for renal dysfunction and progression of IgAN.

Our study also found that there are different degrees of change in mesangial cell proliferation, capillary proliferation, IFTA, and crescent lesions between genotypes; proportions of M1, T2 and C2 of patients with TT genotype were higher than those of non-TT carriers. Several studies have indicated a significant correlation between mesangial hyperplasia, IFTA, crescent and renal dysfunction and progression in IgAN patients (*Bao et al., 2014*; *Xie et al., 2018*). Combined with our findings, IgAN patients carrying *FokI* TT genotype or T allele likely have more serious renal pathological damage. The loss of VDR can lead to renal fibrosis by affecting perirenal inflammation and epithelial-mesenchymal transition (EMT) (*Ito et al., 2013*). *VDR FokI* SNP may influence renal pathology by affecting the quantity or activity of active vitamin D and VDR, which could inhibit EMT, mesangial proliferation and podocyte loss.

Several limitations exist in this study. First, the study was a cross-sectional study with small sample size. Thus, we cannot verify the direct causal relationship between *FokI* polymorphism and renal function. Second, we only evaluated one locus of the *VDR* gene and did not investigate the influence of inheritance of other *VDR* SNPs. Finally, the study did not conduct a dietary questionnaire survey and could not exclude potential effects of diet on indicators related to calcium-phosphorus metabolism. Therefore, large-scale cohort studies should be conducted to confirm our findings and explore additional *VDR* SNPs.

## CONCLUSIONS

IgA nephropathy patients have a higher prevalence of renal dysfunction. Patients with the *FokI* TT genotype are likely to have renal dysfunction, calcium-phosphorus metabolism disorder, mesangial proliferation, IFTA, and crescents in IgAN. *VDR* may be a susceptibility gene for renal dysfunction in IgAN, and *VDR* SNP is closely related to clinical symptoms and pathological damage in IgAN patients.

## ACKNOWLEDGEMENTS

We thank Christina Croney, PhD, from Liwen Bianji, Edanz Group China for editing a draft of this manuscript.

### Funding

This work was supported by the Scientific Research and Technological Development Program of Guangxi (No. GuiKeGong 1598011-6), "Medical Excellence Award" Funded by the Creative Research Development Grant from the First Affiliated Hospital of Guangxi Medical University, the National Natural Science Foundation of China (Nos. 81360111, 81660133) and the Guangxi Natural Science Foundation (2018JJB140279). There was no additional external funding received for this study. The funders had no role in study design, data collection and analysis, decision to publish, or preparation of the manuscript.

### Grant Disclosures

The following grant information was disclosed by the authors:
Scientific Research and Technological Development Program of Guangxi: GuiKeGong 1598011-6.
Creative Research Development Grant from the First Affiliated Hospital of Guangxi Medical University.
National Natural Science Foundation of China: 81360111, 81660133.
Guangxi Natural Science Foundation: 2018JJB140279.

### Competing Interests

The authors declare that they have no competing interests.

### Author Contributions

- Man-Qiu Mo conceived and designed the experiments, performed the experiments, analyzed the data, prepared figures and/or tables, authored or reviewed drafts of the paper, approved the final draft.
- Ling Pan conceived and designed the experiments, performed the experiments, authored or reviewed drafts of the paper, approved the final draft.
- Lin Tan contributed reagents/materials/analysis tools, prepared figures and/or tables.
- Ling Jiang analyzed the data.
- Yong-Qing Pan prepared figures and/or tables.
- Fu-Ji Li contributed reagents/materials/analysis tools.
- Zhen-Hua Yang conceived and designed the experiments, authored or reviewed drafts of the paper, approved the final draft.
- Yun-Hua Liao conceived and designed the experiments, contributed reagents/materials/analysis tools, authored or reviewed drafts of the paper, approved the final draft.

## Human Ethics

The following information was supplied relating to ethical approvals (i.e., approving body and any reference numbers):

The First Affiliated Hospital of Guangxi Medical University granted Ethical approval to carry out the study within its facilities, Ethical Application Ref: 2019(KY-E-006).

## Data Availability

The raw measurements are available in a Supplemental File.

## Supplemental Information

Supplemental information for this article can be found online at http://dx.doi.org/10.7717/peerj.7092#supplemental-information.

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
