# Peer review of "Association between VDR gene FokI polymorphism and renal function in patients with IgA nephropathy"

_PeerJ, doi:10.7717/peerj.7092_

## Round 0.1 · original submission · Major Revisions

Although the manuscript is interesting, there are some concerns about the experimental design, results and discussion section that should be addressed.

Reviewer 1 ·

Basic reporting

1. I would advise the authors to consult a native English speaker to strengthen the manuscript, as there were some grammatical errors;
2. It would be better if the discussion concluded other genes polymorphism in IgA nephropathy, and the clinical influence of the Fokl polymorphism was limited;
3. The authors provide a very large baseline table, with a great number of variables, of which only a small percentage is significantly different.It should be reconstructed;

Experimental design

I suggest that authors improve the description in Subjects part, like what is the meaning by "patients with multiple failure"(Line 70)?

Validity of the findings

1. It is questioned with some data in Table 1, like iPTH. In general, iPTH is abnormal distribution,is not appropriate showed as Mean±SD, and I found that in the detailed data author provide, even in eGFR<60ml/1.73m2, no elevation in iPTH was founded, which is not assistant in clinics;
2.In clinical grouping (Line 85-88), I would suggest the authors clearly state the GFR equation used in this study;
3. In discussion part, I would advise the author to potential reasons for the outcomes with more detailed and newly literature reviews;
4. Althought the authors provide a new gene polymorphism, but no further relationship with clincal dicisions were discussed, and also its advantage and disadvantage with other knowned gens.

Additional comments

I read with interest this manuscript describing a promising gene polymorphism in IgA nephropathy. As described in this manuscript, TT genotype were independent risk factors for renal dysfunction with a dramatic increase. However, the authors do not clearly describe the rationale and also its potential molecular mechanisms and applying prospect.

Reviewer 2 ·

Basic reporting

Man-Qiu Mo et all conducted cross-sectional study in order to investigate the association between VDR Fok I gene polymorphisms and renal function in IgAN patients.

Experimental design

The author conducted cross-sectional study in patients with IgA nephropathy.

Validity of the findings

The results of study is that Fok I TT type have an increased risk of renal dysfunction.

Additional comments

I think this study is meaningful for nephrologist and the results of this study is interesting. On the other hand, this study contains several problems in study design, method, and result.

[Major problems]
1. the author concluded that FokI TT type is the risk factor of renal failure in IgAN patients. However, some former studies reported that other FokI type, CC or CT, is associated with kidney worse kidney function. Is there any reason these differences?
Clin Interv Aging. 2017; 12: 1673–1679.

2. Table 3 showed linear regression analysis for renal function (sCr or eGFR). I think this regression analysis included too many factors to analyze. It is necessary to select adequate factors for regression analysis in this study. In addition, is it appropriate that in regression analysis for sCr, eGFR is selected for explanatory variable?. I think as sCr and eGFR are same meaning, this analysis is inappropriately. In addition, age and age>40 contains multicollinearity.

[Minor problems]
1. PTH is not usually regarded as normal distribution variables. It is necessary to check normality of PTH in this study and correct table 1 appropriately.

2. Is there any information about hematuria?

---

## Round 0.2 · accepted · Accept

The authors have satisfactorily addressed the issues raised by the reviewers.

Reviewer 2 ·

Basic reporting

Man-Qiu Mo et all conducted cross-sectional study in order to investigate the association between VDR Fok I gene polymorphisms and renal function in IgAN patients

Experimental design

The author conducted cross-sectional study in patients with IgAN.

Validity of the findings

The results of study is that FokI TT type have an increased risk of renal dysfunction.

Additional comments

I think this study is meaningful for nephrologist and the results of this study is interesting. The revised manuscript is well corrected and worth publishing.